Morphological, molecular and 3D synchrotron X-ray tomographic characterizations of Helicascus satunensis sp. nov., a novel mangrove fungus

Preedanon Sita 1
Klaysuban Anupong 1
Suetrong Satinee 1
Pracharoen Oraphin 2
Promchoo Waratthaya 2
Sangtiean Tanuwong 2
Rojviriya Catleya 3
Sakayaroj Jariya jsakayaroj@gmail.com 1 4
1 National Biobank of Thailand, National Center for Genetic Engineering and Biotechnology (BIOTEC), National Science and Technology Development Agency (NSTDA) , Khlong Nueng, Khlong Luang , Pathum Thani , Thailand
2 Department of Marine and Coastal Resources, Ministry of Natural Resources and Environment , Laksi , Bangkok , Thailand
3 Synchrotron Light Research Institute (Public Organization) , Nakhon Ratchasima , Thailand
4 School of Science, Walailak University , Nakhon Si Thammarat , Thailand
Robillard Tony
Electronic publication date: 2024 Nov 8
Publication date: 2024
Volume: 12
Electronic Location ID: e18341
Received 2024 Jul 17; Accepted 2024 Sep 26
Copyright: ©2024 Preedanon et al.
Copyright year: 2024
Copyright holder: Preedanon et al.
License: This is an open access article distributed under the terms of the Creative Commons Attribution License, which permits unrestricted use, distribution, reproduction and adaptation in any medium and for any purpose provided that it is properly attributed. For attribution, the original author(s), title, publication source (PeerJ) and either DOI or URL of the article must be cited.
License URL: https://creativecommons.org/licenses/by/4.0/

Keywords: Ascomycota, Marine fungi, 3D synchrotron, Helicascus

Funding: BIOTEC and Department of Marine and Coastal Resources (DMCR) Ministry of Natural Resources and Environment This work was supported by the collaborative project between BIOTEC and Department of Marine and Coastal Resources (DMCR), Ministry of Natural Resources and Environment under a project: Marine Microbes from National Reserves: Alternative Ways for Biotechnological Utilization and Conservation. The funders had no role in study design, data collection and analysis, decision to publish, or preparation of the manuscript.

==============================
A new species of Helicascus satunensis sp. nov. was collected on mature dead fruits of the Nypa palm in Satun Province, southern Thailand. Its morphological characteristics are similar to those of the genus Helicascus. Recently, a genus Helicascus with three species from marine habitats worldwide was studied. The morphology of this fungus was investigated and combined with multigene sequence analyzes of small subunit (SSU), large subunit (LSU), internal transcribed spacer (ITS) ribosomal DNA, translation elongation factor 1-alpha (TEF-1α) and RNA polymerase II (RPB2) genes. Morphologically, H. satunensis sp. nov. is characterized by semi-immersed, lenticular ascomata, multilocules, a bitunicate ascus and smooth, obovoid, dark brown ascospores that are one-septate and unequally two-celled. In addition, 3D visualization using synchrotron X-ray tomography was performed to investigate the interaction between fruiting body and substrata. Molecular phylogeny with multigene revealed that H. satunensis sp. nov. belongs to the family Morosphaeriaceae, order Pleosporales, class Dothideomycetes. Furthermore, H. satunensis sp. nov. forms a well-supported clade with Helicascus species described from marine habitats. Based on the unique morphological and molecular evidence, we propose this fungus, H. satunensis sp. nov., as a new species for Helicascus.

Introduction

Kohlmeyer (1969) described the distinct marine ascomycete Helicascus obtained from dead prop roots of the mangrove Rhizophora mangle. A type species, Helicascus kanaloanus, is characterized by an immersed ascostroma composed of multilocules that share a common periphysate ostiole lying under pseudostromatic tissues. The asci are subcylindrical bitunicate and pediculated, and the endoascus is usually coiled at the base. The ascospores are brown to dark brown at maturity and are frequently asymmetrically two-celled with a mucilaginous sheath in this species (Kohlmeyer, 1969).

Since 1991, a number of Helicascus-like species have been described from freshwater and marine habitats based on their common morphological characteristics and DNA sequences (Hyde, 1991; Zhang et al., 2013; Zhang et al., 2014; Zhang et al., 2015; Luo et al., 2016; Preedanon et al., 2017; Zhang et al., 2024). However, two new genera, Aquihelicascus and Neohelicascus, were excluded from the genus Helicascus due to their morphological and multigene phylogeny (Dong et al., 2020). Aquihelicascus was established to accommodate one new combination and two new species. Neohelicascus was introduced to accommodate one new species and seven new combinations (Dong et al., 2020).

Recently, three marine species in the genus Helicascus were identified, H. kanaloanus, H. nypae and H. mangrovei, based on morphological and molecular data (Kohlmeyer, 1969; Hyde, 1991; Preedanon et al., 2017). Helicascus nypae was found on Nypa palm fronds from Southeast Asia. It is characterized as having ascomata with immersed multilocules with a single common central ostiole, bitunicate asci with a long, narrow and coiled endoascus, and unequally two-celled, verrucose ascospores surrounded by a gelatinous sheath (Hyde, 1991). Subsequently, Preedanon et al. (2017) reported H. mangrovei obtained from decaying mangrove wood in Thailand. The unique morphological characteristics of H. mangrovei include multilocular ascomata semi-immersed under a thick clypeus that forms pseudostromata, clavate pedicellate asci in a hamathecium of cellular pseudoparaphyses, dark brown color at maturity, and unequally two-celled ascospores with one apiculate end.

The aim of this study is to report on novel ascomycete found in Thai mangrove habitats. The microscopic morphology of the fungal fruiting bodies and host tissues was visualized in three dimensions using synchrotron radiation X-ray tomography, which enables high-resolution and non-invasive visualization of internal features without the need for serial sections and staining reagents. This capability is simply unattainable with conventional characterization tools (Friis et al., 2014; Sena et al., 2022; Becher, Sheppard & Grunwaldt, 2023). Furthermore, we provide the molecular phylogeny of the combined SSU, LSU, ITS rDNA, TEF-1α and RPB2 sequences to confirm their taxonomic position.

Materials and Methods

Sample collection, isolation, morphological examination, and materials availability

Mature dead Nypa palm fruits were collected from mangroves at Mangrove Forest Resource Development Station 36 in Satun Province, southern Thailand (6°54′14.9616″N and 99°41′17.4912″E). Collecting procedure was made as previously described in Preedanon et al. (2017). The collected Nypa palm fruits were placed in a sealed plastic Ziploc bag and brought back to the laboratory for further examination. The specimens were washed with natural seawater in order to remove sediment and other debris then kept in a moist plastic box and incubated at room temperature (approximately 25–28 °C). The samples were examined directly under a stereo-zoom microscope for the presence of H. satunensis sp. nov. Photographic documentation of the sporulating structures was carried out using the Olympus BX51 and Olympus DP21 software (Olympus, Tokyo). The ascomata specimens were fixed with embedding matrix on stage and cutting sections through a cryostat microtome (MEV; SLEE Mainz, Mainz, Germany). The fresh ascomata of H. satunensis sp. nov. were selected for single spore isolation (Vrijmoed, 2000). Spore suspension was diluted and plated on 1.5% seawater corn meal agar (SCMA) medium with the addition of antibiotics (streptomycin sulfate 0.5 g/L, penicillin G 0.5 g/L). The germinated spores were placed onto freshly SCMA medium and incubated at room temperature (approximately 25–28 °C) (Preedanon et al., 2017). Axenic cultures (BCC 83546, BCC 86189, BCC 86190) were then transferred to 1.5% seawater potato dextrose agar (SPDA). Colony characteristics, growth and sporulation were observed and recorded. The type cultures were deposited at the BIOTEC Culture Collection (BCC), Pathum Thani, Thailand. In addition, dried voucher type specimens (BBH 50658, BBH 50659 and BBH 50660) were deposited at BIOTEC Bangkok Herbarium (BBH), Pathum Thani, Thailand.

Three-dimensional synchrotron X-ray tomography

The microstructure of the fungal fruiting bodies and the outer exocarp of Nypa palm fruit was visualized in three dimensions using synchrotron radiation X-ray tomographic microscopy (SRXTM). Prior to the SRXTM experiment, the fresh fungal fruiting bodies samples were fixed with 3% formaldehyde. For tomographic imaging, each sample was placed in a sample holder attached to a brass stub with glue to stabilize the sample during tomography scanning. The SRXTM examination of the samples was carried out at the X-ray tomographic microscopy beamline (BL1.2W: XTM) at the Siam Photon Source Facility, Synchrotron Light Research Institute. The X-ray beam was generated from a 2.2-Tesla multipole wiggler radiation source optimized with a toroidal focusing mirror and filtered with aluminum foils to achieve an average energy of 10 keV. All X-ray projections were acquired with a pixel size of 3.61 µm using an imaging system consisting of a 2X objective lens-coupled microscope (Optique Peter, Lentilly, France), a YAG-Ce scintillator (CRYTUR, Turnov, Czech Republic) and the PCO.edge 5.5 sCMOS camera (Excelitas PCO GmbH, Kelheim, Germany). To enhance fine details of the entire sample, a tomographic volume was reconstructed from enlarged composite projections obtained from multiple scans. Each scan covered 180°with a step of 0.2°, resulting in a dataset. Subsequently, the X-ray projection datasets underwent pre-processing, which included flat-field correction, beam intensity normalization and image stitching. Tomographic reconstruction was performed using Octopus Reconstruction software (TESCAN, Ghent, Belgium). The resulting computed tomographic slices were analyzed using ImageJ (http://rsbweb.nih.gov/ij/) and Fiji (http://fiji.sc/Fiji), and the 3D visualization of the tomographic volume was displayed using Drishti software (Limaye, 2012).

DNA extraction, PCR amplification and DNA sequencing

Genomic DNA from fungal mycelia was extracted according to the methods of O’Donnell et al. (1997) and Sakayaroj, Pang & Jones (2011). Ribosomal DNA genes (ITS, SSU, LSU) and protein-coding gene sequences (TEF-1α, RPB2) were amplified by polymerase chain reaction (PCR). The ITS rDNA region was amplified with the primer pair ITS4/ITS5 (White et al., 1990), the SSU region with NS1/NS4 (White et al., 1990), the LSU region with LROR/LR5 (Vilgalys & Hester, 1990), the TEF1-α region with EF1-983F/EF1-2218R (Rehner & Buckley, 2005), the RPB2 region with fRPB2-5F/fRPB2-7cR (Liu, Whelen & Hall, 1999) (Table 1). The component of PCR reaction was performed in a total volume of 50 µL, containing 1 µL DNA template (30–50 ng/ µL), 1 µL of each forward and reverse primers (10 µM), 10 µL master mix of Taq DNA polymerase (Thermo Fisher Scientific Inc., Waltham, MA, USA) and 37 µL of double-distilled water. The PCR conditions for all the genes used were set up using the T100TM thermal cycler (BIO-RAD Laboratories, Inc., California) (Table 1). The PCR products were subsequently purified and sequenced by Macrogen (Seoul, South Korea).

Table 1 PCR primers and amplification profiles used in this study.

DNA region	Primer name	Amplification profile	Reference	
		Denaturation	Repeat steps	Extension		
Internal transcribed spacer rDNA (ITS)	ITS5 ITS4	94 °C (2 min)	35 cycles
94 °C (1 min)
54 °C (1 min)
72 °C (2 min)	72 °C (10 min)	White et al. (1990)	
18S small subunit rDNA (SSU)	NS1 NS4	94 °C
(2 min)	35 cycles
94 °C (1 min)
55 °C (1 min)
72 °C (2 min)	72 °C
(10 min)	White et al. (1990)	
28S large subunit rDNA (LSU)	LROR LR5	94 °C
(2 min)	35 cycles
94 °C (1 min)
55 °C (1.5 min)
72 °C (2.5 min)	72 °C
(10 min)	Vilgalys & Hester (1990)	
Translation elongation factor 1-alpha (TEF 1-α)	EF1-983F EF1-2218R	95 °C
(2 min)	35 cycles
95 °C (1 min)
54 °C (1 min)
72 °C (2 min)	72 °C
(10 min)	Rehner & Buckley (2005)	
RNA polymerase II second largest subunit (RPB2)	fRPB2-5F fRPB2-7cR	94 °C
(3 min)	35 cycles
94 °C (1 min)
54 °C (1 min)
72 °C (1.5 min)	72 °C
(8 min)	Liu, Whelen & Hall (1999)	

Table 2 Taxa and sequences database accession numbers used in this study.

Newly generated sequences are indicated in bold.

Taxon	Strain	GenBank accession no.	
		LSU
rDNA	SSU
rDNA	ITS
rDNA	TEF-1α	RPB2	
Aquihelicascus songkhlaensis	MFLUCC 18-1154T	MN913692	–	MT627680	MT954380	–	
Aquihelicascus songkhlaensis	MFLUCC 18-1273	MN913724	MT864319	MT627696	MT954369	MT878464	
Aquihelicascus songkhlaensis	MFLUCC 18-1278	MN913726	MT864318	MT627693	MT954366	MT878458	
Aquihelicascus thalassioideus	MFLUCC 10-0911T	KC886636	KC886637	KC886635	–	–	
Aquihelicascus thalassioideus	MJF 14020-2	KP637165	–	KP637162	–	–	
Aquihelicascus thalassioideus	JCM 17526	AB807558	AB797268	LC014554	AB808534	–	
Aquihelicascus thalassioideus	CBS 110441	AB807557	AB797267	LC014553	AB808533	–	
Aquihelicascus thalassioideus	KUMCC 19-0094	MT627668	–	MT627689	–	–	
Aquihelicascus yunnanensis	MFLUCC 18-1025T	MN913711	MT864292	MT627728	MT954391	–	
Aquilomyces patris	CBS 135661T	KP184041	KP184077	KP184002	–	–	
Aquilomyces patris	CBS 135760	KP184042	KP184078	KP184004	–	–	
Aquilomyces patris	CBS 135662	KP184043	KP184079	KP184003	–	–	
Aquilomyces rebunensis	CBS 139684T	AB807542	AB797252	AB809630	AB808518	–	
Clypeoloculus akitaensis	CBS 139681T	AB807543	AB797253	AB809631	AB808519	–	
Clypeoloculus hirosakiensis	CBS 139682T	AB807550	AB797260	AB809638	AB808526	–	
Clypeoloculus microsporus	CBS 139683T	AB807535	AB797245	AB811451	AB808510	–	
Clypeoloculus towadaensis	CBS 139685T	AB807549	AB797259	AB809637	AB808525	–	
Didymella fucicola	JK 2932	EF177852	–	EF192138	–	–	
Falciformispora lignatilis	BCC 21118	GU371827	GU371835	–	GU371820	–	
Halojulella avicenniae	BCC 18422	GU371823	GU371831	–	GU371816	GU371787	
Halojulella avicenniae	BCC 20173	GU371822	GU371830	–	GU371815	GU371786	
Halojulella avicenniae	JK 5326A	GU479790	GU479756	–	–	–	
Helicascus kanaloanus	A 237	–	AF053729	–	–	–	
Helicascus kanaloanus	ATCC 18591	KX639748	KX639744	KX957961	KX639756	KX639752	
Helicascus mangrovei	BCC 68258T	KX639745	KX639741	KX957958	KX639753	KX639749	
Helicascus mangrovei	BCC 68260	KX639746	KX639742	KX957959	KX639754	KX639750	
Helicascus mangrovei	BCC 74471	KX639747	KX639743	KX957960	KX639755	KX639751	
Helicascus nypae	BCC 36751	GU479788	GU479754	–	GU479854	GU479826	
Helicascus nypae	BCC 36752	GU479789	GU479755	–	GU479855	GU479827	
Helicascus satunensis	BCC 83546 T	PP866393	PP873998	PP873995	PP915719	PP915722	
Helicascus satunensis	BCC 86189	PP866394	PP873999	PP873996	PP915720	-	
Helicascus satunensis	BCC 86190	PP866395	PP874000	PP873997	PP915721	PP915723	
Leptosphaeria maculans	AFTOL ID-277	DQ470946	DQ470993	–	DQ471062	DQ470894	
Massarina igniaria	CBS 845.96	DQ810223	DQ813511	–	–	–	
Microvesuvius unicellularis	AD 291626	OQ799383	–	OQ799384	OQ866586	–	
Microvesuvius unicellularis	AD 291633T	OQ799391	–	OQ799382	OQ866585	–	
Montagnula opulenta	AFTOL ID-1734	DQ678086	AF164370	–	–	DQ677984	
Morosphaeria muthupetensis	PUFD 87T	MF614796	MF614797	MF614795	MF614798	–	
Morosphaeria ramunculicola	BCC 18404	GQ925853	GQ925838	–	–	–	
Morosphaeria ramunculicola	BCC 18405	GQ925854	GQ925839	–	–	–	
Morosphaeria ramunculicola	JK 5304B	GU479794	GU479760	–	–	GU479831	
Morosphaeria ramunculicola	KH 220	AB807554	AB797264	–	AB808530	–	
Morosphaeria velatispora	BCC 17059	GQ925852	GQ925841	–	–	–	
Morosphaeria velatispora	NBRC 107812	AB807556	AB797266	LC014572	AB808532	–	
Neohelicascus aegyptiacus	MFLU 12-0060T	KC894853	KC894852	–	–	–	
Neohelicascus aquaticus	KUMCC 19-0107	MT627662	MT864314	MT627719	MT954384	–	
Neohelicascus aquaticus	KUMCC 17-0145	MG356477	MG356487	MG356479	MG372317	–	
Neohelicascus aquaticus	MFLUCC 17-2300	MG356478	–	MG356480	MG372316	–	
Neohelicascus aquaticus	MFLUCC 10-0918T	KC886640	KC886638	KC886639	MT954384	–	
Neohelicascus aquaticus	MAFF 243866	AB807532	AB797242	AB809627	AB808507	–	
Neohelicascus chiangraiensis	MFLUCC 13-0883T	KU900585	KU900587	KU900583	KX455849	–	
Neohelicascus elaterascus	MAFF 243867	AB807533	AB797243	AB809626	AB808508	–	
Neohelicascus elaterascus	CBS 139689	LC014608	LC014603	LC014552	LC014613	–	
Neohelicascus elaterascus	MFLUCC 18-0985	MT627658	MT864335	MT627735	–	–	
Neohelicascus elaterascus	MFLUCC 18-0993	MT627659	MT864333	MT627730	–	–	
Neohelicascus elaterascus	HKUCC 7769	AY787934	AF053727	–	–	–	
Neohelicascus gallicus	BJFUCC 200228	KM924831	–	KM924833	–	–	
Neohelicascus gallicus	CBS 123118	KM924832	–	–	–	–	
Neohelicascus gallicus	BJFUCC 200224	KM924830	–	–	–	–	
Neohelicascus griseofavus	MFLUCC 16-0869T	OP377964	OP378041	OP377878	OP473055	–	
Neohelicascus submersus	MFLU 20-0436T	MT627656	MT864340	MT627742	–	–	
Neohelicascus unilocularis	MJF 14020T	KP637166	–	KP637163	–	–	
Neohelicascus unilocularis	MJF 14020-1	KP637167	–	KP637164	–	–	
Neohelicascus uniseptatus	MFLUCC 15-0057T	KU900584	–	KU900582	KX455850	–	
Paradendryphiella arenariae	AFTOL ID-995T	DQ470971	DQ471022	–	DQ677890	DQ470924	
Parastagonospora avenae	AFTOL ID-280	AY544684	AY544725	–	DQ677885	DQ677941	
Phaeodothis winteri	AFTOL ID-1590	DQ678073	DQ678021	–	DQ677917	DQ677970	
Phaeosphaeria eustoma	AFTOL ID-1570	DQ678063	DQ678011	–	DQ677906	DQ677959	
Platychora ulmi	CBS 361.52	EF114702	EF114726	–	–	–	
Plenodomus biglobosus	CBS 303.51	GU301826	–	–	GU349010	–	
Setoseptoria arundinacea	CBS 619.86	DQ813509	DQ813513	–	–	–	
Stemphylium vesicarium	CBS 191.86T	DQ247804	DQ247812	–	DQ471090	DQ247794	
Trematosphaeria pertusa	CBS 122371	FJ201992	FJ201993	–	–	–	
Outgroup	 	 	 	 	 	 	
Lophiostoma macrostomum	JCM 13544	AB619010	AB618691	JN942961	LC001751	JN993491	
Sigarispora arundinis	JCM 13550	AB618998	AB618679	JN942964	LC001737	JN993482	
Notes.

T Ex-type strain

Phylogenetic analyses

Multiple sequence alignments were analyzed with the closely matched sequences obtained from GenBank (Table 2) according to Jones et al. (2015), Maharachchikumbura et al. (2016) and Hongsanan et al. (2017), Dong et al. (2020), Yang et al. (2023a) and Yang et al. (2023b). The newly generated sequences from this study are listed in Table 2. The nucleotide sequences were assembled and aligned using BioEdit 7.2.5 (Hall, 1999) and Muscle 3.8.31 (Edgar, 2004). Specifically, NCBI blast searches were used to determine sequence similarity to sequences published in the GenBank database. Phylogenetic analyses of the combined SSU, LSU, ITS rDNA, TEF-1α and RPB2 sequences were performed using maximum likelihood (ML) and Bayesian algorithms. Maximum likelihood (ML) analysis was evaluated in RAxMLHPC2 on XSEDE (Stamatakis, 2014) via the CIPRES Science Gateway platform (Miller, Pfeiffer & Schwartz, 2010) under the GTR + GAMMA model with BFGS method to optimize the GTR rate parameters. Finally, Bayesian posterior probabilities of branches were performed using MrBayes 3.2.6 (Ronquist et al., 2012), with the best-fitting model (GTR+I+G) selected by AIC in MrModeltest 2.2 (Nylander, 2004), which was tested with hierarchical likelihood ratios (hLRTs). Three million generations were run in four Markov chains and a sample was drawn every 100 generations with a burn-in value of 3,000 sampled trees. Finally, the consensus tree was displayed using the interactive Tree Of Life (iTOL) (Letunic & Bork, 2021) and adjusted in Adobe Photoshop 2020. All sequences obtained in this study were submitted to GenBank, and the typification were published in the MycoBank database (Crous et al., 2004). The resulting alignments were submitted to TreeBASE (submission numbers: 31389).

Nomenclature

The electronic version of this article in Portable Document Format (PDF) will represent a published work according to the International Code of Nomenclature for algae, fungi, and plants, and hence the new names contained in the electronic version are effectively published under that Code from the electronic edition alone, so there is no longer any need to provide printed copies. In addition, new names contained in this work have been submitted to MycoBank from where they will be made available to the Global Names Index. The unique MycoBank number can be resolved and the associated information viewed through any standard web browser by appending the MycoBank number contained in this publication (MB 854336) to the prefix http://www.mycobank.org/MB/. The online version of this work is archived and available from the following digital repositories: PeerJ, PubMed Central SCIE, and CLOCKSS.

Results

Taxonomy

Helicascus satunensis Preedanon, Suetrong & Sakay., sp. nov.Fig. 1.	
MycoBank (MB# 854336)	
GenBank (SSU rDNA=PP873998, LSU rDNA=PP866393, ITS rDNA=PP873995, TEF-1α=PP915719, RPB2=PP915722)	

Type: THAILAND, Satun Province, mangrove forests, on a piece of dead palm (Nypa fruticans) fruit, 22 December 2016, S. Preedanon, A. Klaysuban, O. Pracharoen & J. Sakayaroj, BBH 50658, holotypus, cultura dessicata, (holotype designated here) (BIOTEC Bangkok Herbarium, Pathum Thani, Thailand).

Ex-type culture: MCR 00699 (BCC 83546) (BIOTEC Culture Collection, Pathum Thani, Thailand).

Etymology: ‘satunensis’ referring to the collecting location, Satun Province, southern Thailand, where the fungus was collected.

Sexual morph: Ascomata 1,000–2, 400 × 160–280 µm, semi-immersed, lenticular ascomata, 3–4 locules, dark brown to black, carbonaceous, solitary (Fig. 1).

Three-dimensional synchrotron X-ray tomographic analysis reveals that the fungal tissues growing in the outer exocarp of Nypa palm fruits, enclosing 3–4 locules with flattened base, horizontally arranged under the pseudostroma. Cellular, numerous, persistent, hyaline pseudoparaphyses.

Asci 475–642. 5 × 62.5–80 µm, 8-spored, bitunicate asci, cylindrical, thick-walled, short hook pedunculate, with an ocular chamber. Ascospores 22.5–25 × 5–8.75 µm, unequally two-celled, smooth, dark-brown, and slightly constricted at the septum, thick-walled (Fig. 1).

Habitat and distribution: mangrove forests, Satun Province, southern Thailand.

Asexual morph: Undetermined

Culture characteristics: Ascospores germinated on SCMA after 1–2 days, colonial grown on SPDA attaining 2–3 cm in diameter after 60 days incubation at room temperature (approximately 25–28 °C), dense, circular, irregular and grey (7D1) with orange patches (7C5) in the center, white (7A1) at the edge; dark brown at reverse side. Colour codes in the fungal description follow “Methuen Handbook of Colour” (Kornerup & Wanscher, 1978).

Phylogenetic analyses

The phylogenetic relationships of the Pleosporales, Dothideomycetes were reconstructed using the combined five-gene dataset (SSU, LSU, ITS rDNA, TEF-1α, RPB2), with Lophiostoma macrostomum JCM 13544 and Sigarispora arundinis JCM 13550 as the outgroups (Table 2). The alignment of 74 taxa comprised 5,844 base pairs (1,342 for SSU, 1,358 for LSU, 1,192 for ITS, 968 for TEF-1α and 984 for RPB2). Total 3,825 characters were constant; 1,570 characters were parsimony-informative and 449 variable characters were parsimony-uninformative. Phylogenetic analyses showed that our novel species (in bold) belongs to the Morosphaeriaceae. Although the topology of the BI tree and the exhibited ML tree are comparable, the BI tree is not shown. The phylogenetic trees representing the unique position of other species in the marine habitat were deposited in MycoBank. Significant ML bootstrap values (≥50%) and Bayesian posterior probabilities (≥0.95) are indicated in the phylogenetic tree (Fig. 2).

Figure 1 Morphological features of Helicascus satunensis sp. nov.

(A) Carbonaceous ascomata on the exocarp of Nypa palm fruit. (B) The 3D visualization of the ascoma by X-ray tomography (arrow). (C) Vertical section through an ascoma. (D) Section through an ascoma using 3D visualization. (E) Dead Nypa palm fruits. (F–G) Obverse and reverse views of cultures grown on SPDA after 60 days. (H–J) Subcylindrical bitunicate asci. (K–M) Ascospores. (N) Pseudoparaphyses (arrow). Scale bars A = 1 mm, C =300 µm, H–J = 100 µm, K–M = 10 µm, N = 20 µm.

Figure 2 The RAxML phylogenetic tree of Helicascus satunensis (BCC 83546, BCC 86189, BCC 86190) resulted from the combined of LSU, SSU, ITS, TEF-1α and RPB2 sequences.

Lophiostoma macrostomum and Sigarispora arundinis were used as outgroups. Maximum likelihood (BSML, left) equal to or greater than 50% are shown above each branch. Bayesian posterior probabilities (BYPP, right) equal to or greater than 0.95 are shown below each branch. The nodes that are strongly supported by bootstrap proportions (100%) and posterior probabilities (1.00) are shown in a thicker line. Abbreviations: T = ex-type. Novel species is demonstrated in bold.

In the multigene phylogenetic analysis, the Morosphaeriaceae are divided into subclades representing the seven accepted genera including Neohelicascus, Aquihelicascus, Helicascus, Morosphaeria, Clypeoloculus, Microvesuvius, and Aquilomyces. Our fungal strains (BCC 83546, BCC 86189 and BCC 86190) are monophyletic and well placed in the Morosphaeriaceae with robust bootstrap and Bayesian supports. They are phylogenetically distinct from the type species H. kanaloanus and form sister subclades with H. nypae and H. mangrovei (Fig. 2). Within the Helicascus subclade, we compared the base substitutions of our new fungus with the type species H. kanaloanus. The result shows the base substitutions at several sites of SSU (960/969 = 99.0% similarity), LSU (803/853 = 94.1% similarity), ITS rDNA (448/714 = 62.7% similarity), TEF-1 α (872/933 = 93.4% similarity) and RPB2 (793/899 = 88.2% similarity) (Table 3).

Discussion

Taxonomy

Jones et al. (2022) reported 1,900 marine fungi in 769 genera that have evolved for marine life as saprobes, parasites and endophytes. Devadatha et al. (2021) and Zhang et al. (2024) reported that many new taxa have been described from mangrove trees and salt marsh plants. Among these host plants, palms found in mangroves and estuaries, such as Phoenix paludosa, Oncosperma tigillarium, and N. fruticans, harbour a great diversity of fungi (Zhang et al., 2024).

Some of the fungi are found as saprobes on the petioles of N. fruticans: Bacusphaeria nypae, Manglicola guatemaelensis, and Tirisporella baccariana (Jones et al., 1996; Suetrong et al., 2009; Abdel-Wahab et al., 2017). Acuminatispora palmarum, Fasciatispora nypae, Helicascus nypae, Neomorosphaeria mangrovei, Pleurophomopsis nypae, Striatiguttula nypae, and S . phoenicis grow on submerged rachis and petioles of N. fruticans and Ph. paludosa (Hyde et al., 1999; Hyde & Alias, 2000; Loilong et al., 2012; Zhang et al., 2018; Zhang et al., 2019; Zhang et al., 2024). A few fungi, however, were discovered on Nypa fruits: Anthostomella nypae., Fasciatispora spp., and Vaginatispora nypae (Jayasiri et al., 2019; Zhang et al., 2024).

Table 3 Pairwise DNA comparison of H. satunensis sp. nov. with the Helicascus species.

DNA sequence	Number of bases for comparison
(bp)	Helicascus kanaloanus	H. nypae	H. mangrovei	
		Base substitutions	%
similarity	Base substitutions	%
similarity	Base substitutions	%
Similarity	
SSU rDNA	969	9	99.0	14	98.5	9	99.0	
LSU rDNA	853	50	94.1	47	94.5	40	95.3	
ITS rDNA	714	266	62.7	ND	ND	265	62.9	
TEF-1α	933	61	93.4	74	92.0	64	93.1	
RPB2	899	106	88.2	115	87.2	107	88.1	
Notes.

ND Not determined

The genus Helicascus is a distinct marine ascomycete characterized by having a pseudostroma composed of host cells enclosed in fungal hyphae, subcylindrical asci, uniseriate, obovoid, dark brown color at maturity, and asymmetrical ascospores (Kohlmeyer, 1969). Recently, three species have been identified in the genus, namely, H. kanaloanus (type species), H. nypae, and H. mangrovei (Kohlmeyer, 1969; Hyde, 1991; Preedanon et al., 2017). We found a new fungus, H. satunensis, that inhabits the brackish waters of Nypa palm fruit in Satun Province, southern Thailand. Helicascus satunensis shares similar ascostromata with H. kanaloanus and H. nypae in having semi-immersed or immersed, carbonaceous, multilocules in the ascostromata arranged under a black pseudoclypeus, while H. mangrovei does not have separate locules in the ascomata (Table 4).

Table 4 Morphological comparison among species of Helicascus (Kohlmeyer, 1969; Hyde, 1991; Preedanon et al., 2017; Zhang et al., 2024).

	H elicascus kanaloanus	H. nypae	H. mangrovei	H. satunensis sp. nov.	
Pseudostromata					
Size (µm)	600–780 × 1,250–2,750	260–390 × 750–1,500	1,500–1, 750 × 1,500–2,500	1,000–2, 400 × 160–280	
Position on substrata	Immersed	Immersed	Semi-immersed	Semi-immersed	
Locules	Multilocules (3–4 loculi)	Multilocules (3–4 loculi)	Single locule	Multilocules (3–4 loculi)	
Structure	Ampulliform, lenticular, horizontally arranged under a black pseudoclypeus	Lenticular, black, carbonaceous	Lenticular, flattened, carbonaceous, solitary, a locule covered by a pseudoclypeus	Lenticular, black, carbonaceous, solitary	
Asci					
Size (µm)	200–260 × 15–25	192–280 × 14–20	400–412. 5 × 25–30	475–642. 5 × 62.5–80	
Shape	Subcylindrical to oblong clavate, persistent, pedunculate, thick-walled	Subcylindrical, pedunculate	Subcylindrical, pedunculate, thick-walled	Cylindrical, thick-walled	
Endoascus	With an apical apparatus
coiling	With an ocular chamber	With an apical apparatus
coiling	Short hook pedunculate with an ocular chamber	
Pseudoparaphyses	Cellular, numerous, persistent	Cellular, numerous, persistent, anastomosing in a gel	Cellular, numerous, trabeculate, hyaline	Cellular, numerous, persistent, hyaline	
Ascospores					
Size (µm)	30–55 × 17–25	25–35 × 12 –15	40–45 × 18.5–20	22.5–25 × 5–8.75	
Sheath	Present in some collection	Present	Absent	Absent	
Shape	Obovoid, brown, biseriate
one-septate, constricted at the septum, dark-brown at maturity, unequally two-celled	Uniseriate, obovoid, constricted at the septum, brown, sometimes at one or both ends apiculate,
unequally two-celled	Uniseriate, obovoid, unequally two-celled, slightly constricted at the septum, thick-walled
and only one apiculate end,
dark brown at maturity	Constricted at the septum, thick-walled, unequally two-celled, dark brown at maturity	
Ornamentation	Smooth wall	Verrucose wall	Smooth wall	Smooth wall	
Host	Dead mangrove wood	Nypa fruticans, Phoenix paludosa fronds	Dead mangrove wood	Nypa fruticans fruit	
Asexual morph	Undetermined	Pleurophomopsis nypae	Undetermined	Undetermined	

Pseudostroma is a unique taxonomic characteristic of the Morosphaeriaceae at the genus level. Zhang et al. (2015) reported that multilocular pseudostroma are important in delineating species of Helicascus-like species. Members of the Morosphaeriaceae develop somatic hyphae into ascostroma, which subsequently form locules that include the genera Helicascus, Neohelicascus, Aquihelicascus, Morosphaeria, Clypeoloculus, and Aquilomyces (Dong et al., 2020). The coiling and stretching mechanism of the basal endoascus with an ocular chamber are regarded as unique types of asci in the genus Helicascus. Our new fungus H. satunensis shares this type of asci with three other species in the genus (Zhang et al., 2015). All species share the same arrangement of cellular pseudoparaphyses. The ascospores of H. satunensis can be distinguished from those of other species because they are smaller (22.5–25 × 5–8.75 µm) than those of H. kanaloanus (30–55 × 17–25 µm), H. nypae (25–35 × 12–15 µm), and H. mangrovei (40–45 × 18.5–20 µm). Germ pores were not observed in H. satunensis, while they appeared at only one end in H. mangrovei (Preedanon et al., 2017). Moreover, the unequal number of H. satunensis 2-cell cones with constriction of ascospores could be a defined taxonomic marker at the species level in the genus Helicascus.

Molecular phylogeny

Phylogenetic analyses of multigene sequences revealed that Helicascus satunensis forms a well-supported clade within the Morosphaeriaceae, Pleosporales, Dothideomycetes. The Morosphaeriaceae family was established by Suetrong et al. (2009) based on morphological features and strong phylogenetic support. Currently, the family comprises eight genera: Aquihelicascus (Dong et al., 2020), Aquilomyces (Knapp et al., 2015), Clypeoloculus (Tanaka et al., 2015), Helicascus (Kohlmeyer, 1969; Dong et al., 2020), Microvesuvius (Fryar, Réblová & Catcheside, 2023), Morosphaeria (Suetrong et al., 2009), Neohelicascus (Dong et al., 2020), and Neomorosphaeria (Zhang et al., 2024).

Members in the Morosphaeriaceae family are found on submerged dead twigs in freshwater and marine environments. The multigene phylogeny comprising freshwater taxa formed sister clades to the marine fungal lineages. Two new freshwater genera, Aquihelicascus and Neohelicascus, were excluded from the genus Helicascus due to morphological and molecular evidence (Dong et al., 2020). Aquihelicascus was established to accommodate one new combination (A. thalassioideus) and two new species (A. songkhlaensis and A. yunnanensis). Neohelicascus was introduced to accommodate one new species (N. submersus) and seven new combinations (N. elaterascus, N. chiangraiensis, N. unilocularis, N. uniseptatus, N. aegyptiacus, N. gallicus, and N. aquaticus) (Dong et al., 2020). The genera Morosphaeria (M. ramunculicola, M. muthupetensis, M. velatispora) (Suetrong et al., 2009; Devadatha et al., 2018), Neomorosphaeria (Zhang et al., 2024), and Helicascus (H. kanaloanus, H. nypae, H. mangrovei) are predominant saprobic on decaying mangroves and marine substrata, while only Aquilomyces patris is a root endophyte of white poplar (Fryar, Réblová & Catcheside, 2023).

The multigene phylogeny in the present study showed that our new fungus H. satunensis forms a distinct lineage within the genus Helicascus with robust statistical support (100% ML bootstrap and 1.00 Bayesian posterior probability). The DNA sequences of H. satunensis differ from those of H. kanaloanus and other species in terms of the number of nucleotide base substitutions in all the DNA regions, which indicates that these species are different. In conclusion, with its unique morphological and multigene phylogeny, we introduce H. satunensis as a novel mangrove fungus.

Supplemental Information

Supplemental Information 1 Multigene DNA sequence alignment

Supplemental Information 2 SSU sequences for GenBank submission

Supplemental Information 3 LSU sequences for GenBank submission

Supplemental Information 4 ITS sequences for GenBank submission

Supplemental Information 5 TEF sequences for GenBank submission

Supplemental Information 6 RPB2 sequences for GenBank submission

We acknowledge Prof. E.B. Gareth Jones, Prof. Morakot Tanticharoen, Dr. Kanyawim Kirtikara, Dr. Lily Eurwilaichitr, Dr. Somvong Tragoonrung, and Dr. Sissades Tongsima for continued support. Special thanks to Pranom Chumriang for field assistance.

Additional Information and Declarations

Competing Interests

Author Contributions

DNA Deposition

Data Availability

New Species Registration

The authors declare there are no competing interests.

Sita Preedanon conceived and designed the experiments, performed the experiments, analyzed the data, prepared figures and/or tables, authored or reviewed drafts of the article, and approved the final draft.

Anupong Klaysuban conceived and designed the experiments, performed the experiments, authored or reviewed drafts of the article, field assistance, and approved the final draft.

Satinee Suetrong conceived and designed the experiments, performed the experiments, authored or reviewed drafts of the article, and approved the final draft.

Oraphin Pracharoen conceived and designed the experiments, performed the experiments, authored or reviewed drafts of the article, sample preparation, and approved the final draft.

Waratthaya Promchoo conceived and designed the experiments, performed the experiments, authored or reviewed drafts of the article, field assistance, and approved the final draft.

Tanuwong Sangtiean conceived and designed the experiments, performed the experiments, authored or reviewed drafts of the article, field assistance, and approved the final draft.

Catleya Rojviriya conceived and designed the experiments, performed the experiments, analyzed the data, authored or reviewed drafts of the article, and approved the final draft.

Jariya Sakayaroj conceived and designed the experiments, performed the experiments, analyzed the data, prepared figures and/or tables, authored or reviewed drafts of the article, and approved the final draft.

The following information was supplied regarding the deposition of DNA sequences:

The DNA Sequences are available at GenBank: PP866393, PP873998, PP873995, PP915719, PP915722, PP866394, PP873999, PP873996, PP915720, PP866395, PP874000, PP873997, PP915721, PP915723.

The following information was supplied regarding data availability:

MycoBank number contained in this publication MB# 854336. http://www.mycobank.org/MB/.

The Mycobank submission sheet is available in the Supplement File.

The following information was supplied regarding the registration of a newly described species:

Helicascus satunensis Preedanon, Suetrong & Sakay., sp. nov. MycoBank (MB#854336) http://www.mycobank.org/MB/

The type cultures were deposited at the BIOTEC Culture Collection (BCC), Pathum Thani, Thailand. In addition, dried voucher type specimens (BBH 50658, BBH 50659 and BBH 50660) were deposited at BIOTEC Bangkok Herbarium (BBH), Pathum Thani, Thailand.

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
