# Peer review of "Morphological, molecular and 3D synchrotron X-ray tomographic characterizations of Helicascus satunensis sp. nov., a novel mangrove fungus"

_PeerJ, doi:10.7717/peerj.18341_

## Round 0.1 · original submission · Minor Revisions

· Academic Editor

Minor Revisions

Both reviewers agree that this work is worthy of publication, but they also raised a number of relatively minor issues that I encourage you to follow to ameliorate the manuscript.

Reviewer 1 ·

Basic reporting

This work is on the report of a new species of Helicascus using various methods.
It can be considered for a further process after revision.

Experimental design

There are many things to be revised for a further process.

Validity of the findings

No comment

Additional comments

I think that traditional marine fungal species are different from those living in/on plants in seawater environments.
The marine fungal species in this work should be revised to indicate fungi living in/on plants in seawater environments.

Why did you use the synchrotron X-ray imaging in this work, not the non-synchrotron one?

Is the fungus a pathogen of Nypa palm?
Describe its implications with the plant in the Taxonomy.

Please revise the manuscript based on the follows:

95: Synchrotron --> synchrotron
111: Spell out CT
120: PCR reactions: redundancy
123: Macrogen (Seoul, South Korea).
94, 177: Which host tissues? Stem? How old?

Fig. 1A: ascoma --> ascomata; Add arrows to idicate ascomata.
Fig. 1B: arrowhead --> arrow
Fig. 1C: Add descriptions on sectioning to the M&M section.
Fig. 1E: X-ray CT image? arrowhead --> arrow; The arrow indicates only opening on the plant surface, not ostiole.
Fig. 1N: Add arrows to idicate pseudoparaphyses.

Reviewer 2 ·

Basic reporting

The manuscript is written in clear and professional English throughout with relevant literature references. However the authors can consider adding more recent literature references. The article structure, figures and tables are professional and raw data was shared.

Experimental design

This section was well done by the authors however more information is needed to explain how sampling was done and how the samples were transported to the laboratory. Were they contained in Ziploc bags or in a cooler box?

Line 84; Any processing of the sample done prior to examination under the microscope? This will help to improve this section.

Lines 85-87; More information on how the samples were processed before inoculation and incubation conditions to be provided to improve this section

Line 120; Primers used and PCR conditions should be provided

Generally the methods were described with sufficient details and information but need to be improved as stated above to allow replication.

Validity of the findings

All underlying data have been provided and well discussed.
Conclusions are well stated and linked to the research questions

Additional comments

I recommend the authors to have a complete thorough grammar-, spell-, and punctuation-check on the entire document

---

## Round 0.2 · accepted · Accept

· Academic Editor

Accept

Thank you for addressing appropriately all the minor comments and suggestions of the reivewers. I consider that the manuscript can now be accepted for publication.